# A win-win scenario? Employers' responses to HIV in Tanzania: A qualitative study

**Kevin Deane** [1]*, **Joyce Wamoyi** [2], **Samwel Mgunga** [2], **John Changalucha** [2]

**1** Economics Department, Open University, Milton Keynes, United Kingdom, **2** Sexual and Reproductive Health Department, National Institute for Medical Research, Mwanza Research Centre, Mwanza, Tanzania

* kevin.deane@open.ac.uk

**Data Availability Statement:** Our article contains excerpts from the qualitative data we collected and synthesized. At the time that ethical approval was applied for, we did not specify that data would be

## Abstract

Workplaces have been increasingly promoted as key sites for HIV interventions, with cost-benefit analyses employed to demonstrate the financial benefits to employers for implementing workplace HIV programmes. In these analyses, the potential costs of having HIV positive employees are weighed against the costs of the workplace programmes. Despite evidence that shows some firms have saved significant sums of money through these interventions, the general response from the private sector has been limited, with most positive case studies originating from high prevalence settings. This article reports findings from qualitative fieldwork conducted in Tanzania with private and public sector employers that aimed to understand how HIV was addressed in their organisations. Our findings suggest that HIV is not generally a serious issue, and hence HIV interventions are primarily ad-hoc with few formal HIV workplace programmes. We also found that in cases where compulsory testing programmes were implemented, employees did not turn up for testing and thus lost access to employment. Our findings suggest that relying on workplace programmes in lower prevalence settings is no substitute for investment in public health systems. Employer interventions should emphasise education and awareness, condom distribution and the promotion and provision of self-testing kits.

## Introduction

Adherence to Antiretroviral Therapy (ART) enables People Living With HIV (PLWHIV) to live a normal life span [1, 2], as well as significantly reducing the likelihood that they will transmit the virus. These dual benefits, which address both treatment and prevention concerns, have positioned expanding access to and uptake of HIV testing and treatment services as central to the current 95-95-95 UNAIDS agenda (by 2030, 95% of PLWHIV know their status, 95% of PLWHIV who know their status are receiving antiretroviral therapy, and 95% of those on treatment have a supressed viral load) [3]. Whilst there has been a concerted global effort to fund access to treatment in sub-Saharan African countries, as well as significant progress, the roll-out of testing and treatment programmes in that context faces several challenges, including fragile and under-funded public health systems and concerns about state capacity. To that end, the workplace has been increasingly viewed as a conducive site for HIV related interventions for both treatment and prevention activities to complement public sector initiatives, with

made available in a public repository. Therefore doing so would not be in line with the approved ethical application and the information that was provided to participants about the study and how their data would be used. Making the raw data publicly available would be a serious ethical breach in relation to the rights of the study participants. Further, given the well-documented methodological challenges when researching with (and especially recruiting) elite participants, we suspect that the study would not have been possible if participants had been informed that their data would be made publicly available. In case there is special interest in our data, access requests can be made to the Tanzania National Health Research Ethics Committee at ethics@nimr. or.tz.

**Funding:** This study was supported by the Leverhulme Trust in the form of a grant (SG153365) awarded to KD and JW. No additional external funding was received for this study. The funders had no role in study design, data collection and analysis, decision to publish, or preparation of the manuscript.

**Competing interests:** The authors have declared that no competing interests exist.

the International Labour Organisation (ILO) noting that "challenges such as the persistence of HIV-related stigma and discrimination, gender inequalities, and lack of integration of HIV into broader health and development plans are being addressed by effective and robust programmes in the African workplaces, which are now leading entry points for the response to HIV and AIDS" [4].

Across the continent, national policy frameworks govern the response to HIV in the workplace with a focus on the rights of employees to non-discrimination based on their HIV status, confidentiality, protection against pre-employment screening or enforced testing within the workplace, and the need for employers to support access to services [5–7]. However, more recently, arguments for an expanded role for employers are rooted in a cost-benefit approach that seeks to quantify the existence of a business case for a more comprehensive degree of intervention (or not) [8–16]. In this approach, the potential costs associated with having HIV positive employees are weighed against the costs of implementing an HIV workplace intervention or programme. The potential costs include impacts on labour productivity, support with treatment costs, increased number of days off due to illness or visits to healthcare facilities, increased health insurance costs, higher worker turnover leading to increased recruitment and training costs, and a more general negative impact on workplace morale [8, 14, 17]. These costs are then compared to the costs of the workplace interventions themselves, which can include (but are often quite diverse in design) staff training (including resources for trainers and time off for workers), access to HIV testing and treatment services, condom promotion and distribution, peer-to-peer support, support with the financing of treatment, and monitoring and evaluation activities that track HIV prevalence within the workforce (and in some cases the outcomes of specific interventions) [17]. This method thus establishes the net costs that employers can avoid (i.e. the benefits) if the workplace intervention is implemented, with evidence from a number of studies demonstrating that the net costs avoided are often significant [8, 11, 14]. The result is a 'win-win scenario' in which HIV workplace programmes can have positive health outcomes for workers and save large sums of money for employers [15]. This case for intervention is further supported by recent evidence that demonstrates that HIV workers who are able to access and adhere to treatment are as productive as other workers [18]. An added benefit is that these interventions do not require legislation or monitoring by the state (which is often regarded with suspicion by external donors funding HIV programming), as private sector firms, which are the focus of the literature, will respond to the epidemic in a supportive way as it is financially beneficial for them to do so. Therefore, policy makers are directed to focus on ensuring that firms understand the business case rather than the expansion (and enforcement) of national and sectoral HIV policies.

However, whilst there are some prominent examples of successful workplace programmes that have reportedly saved employers vast amount of money [11, 13–15, 19], the general consensus in the literature is that the response from employers, and especially those in the private sector, has been weak [20–22]. This is due to a number of issues (see [23] for a full review) that include the questionable existence of the business case when other factors are accounted for (firm size, sector and skill levels of employees), and alternative strategies that employers can deploy to 'shift the burden' of these costs, such as pre-employment screening, managing HIV positive employees out of the organisation and reducing in-work health benefits [24–27]. The empirical evidence, which primarily originates from a narrow range of multinational/blue chip firms in South Africa [16], is less compelling once evidence in other sectors and countries is accounted for, and especially with respect to whether workplace programmes have the intended impact of reducing new HIV infections and HIV-related morbidity and mortality [4]. Indeed, it is not even clear whether workplace programmes are effective [20]. There are also other factors beyond the business case that also influence how employers respond, such as

internal HIV 'champions' who have personal experiences of HIV, and corporate social responsibility requirements for organisations linked with the epidemic (for example the mining sector) [28, 29], suggesting a more nuanced picture and alternative modes of decision making that do not follow the business case logic. Further, there has been little research on how employers actually respond. Most studies that take the business case approach utilise human resources records to calculate the costs outlined above, whilst the evidence regarding the alternative ways that employers seek to shift the burden is largely anecdotal [25] and is not incorporated into the standard cost-benefit approach. The small body of literature that exists emphasises the challenges of implementing workplace programmes from the perspective of employees and lower level managers highlighting issues such as confidentiality and HIV stigma as key factors that influence the operational success of workplace programmes [30].

This article addresses this lack of research by reporting findings from a qualitative study that explored how employers (both private sector and public sector) viewed and responded to the issue of HIV in the workplace in the Tanzanian context. The research aimed to understand the impact of the epidemic on private and public sector organisations, and how employers respond to the issue of HIV.

## Methods

### Study setting

This study was conducted in Mwanza city, Tanzania. Most of the existing literature focuses on contexts in which there is a high background prevalence of HIV, as well as HIV rates of HIV within the workforce, and as such employers have been forced to confront the issue [16, 23]. However, in Mwanza region, the background HIV prevalence is around 6.8% [31], creating a situation in which this external imperative to intervene may not have been as keenly felt. This means, perhaps counter intuitively, that it is vital to understand how employers address HIV in those contexts where HIV may not be considered a serious issue, particularly as lower background prevalence rates may put HIV positive employees in a more vulnerable position if they can be more easily replaced [23]. Mwanza City was selected as a study site as it is a significant source of formal employment (both private and public sector) in northern Tanzania and has a comparable HIV prevalence rate to other major urban areas in the region (such as Kigali, Kamapala and Nairobi) [32–34]. In the Tanzanian context, progress towards the UNAIDS 95-95-95 targets is uneven, with only 60.0% of PLWHIV knowing their status [35]. However, for those who do know their status, 93.6% are on ART, and of those, 87% have viral load suppression (*ibid*). This emphasises the need to expand access to and uptake of HIV testing, often included as a core component of the workplace programmes discussed above, to accelerate progress towards UN targets.

### Sampling approach and characteristics of participants

Potential participants were initially identified through brainstorming sessions with the research team and other contacts within the National Institution for Medical Research, Mwanza centre. This included discussions about the most important sectors in Mwanza (for example Manufacturing, Construction, Wholesale and Retail Trade, Transportation and Storage, Accommodation and food services, Information and Communication, Financial and insurance activities, Professional scientific and technical activities [36], businesses that would likely be locally owned compared to businesses that would be managed on a regional basis but owned either by pan-Tanzanian or International firms, and the key public institutions that employ large numbers of workers. This list served as a starting point for approaching potential participants, though it was added to as the fieldwork progressed. Participants were contacted

in a range of ways, such as through personal networks of the research community in Tanzania, the personal networks of friends of the research community, institutional relationships (such as local public financial institutions or private sector firms that had engaged commercially with the host research institution) and snowballing through participants. Participants were contacted via phone or via appointments made in person at their workplace. None of the participants were known to the study team prior to the data collection. Participants were included if they were the owner of a private sector organisation, held a regional management position in a national or international private sector organisation, or were the regional officer for a public sector organisation, based in Mwanza city. The regional/zonal management roles are standardised organisational roles in the Tanzanian context (similar to roles such as Chief Executive Officer), meaning that the inclusion criteria applied to specific individuals in each organisation that we approached. Throughout the data collection period, a total of 36 potential participants were approached, but due to a range of issues related to the sensitivity of the research, wariness of the research process amongst a population that is very rarely approached in the Tanzanian context, and an increasingly hostile socioeconomic environment in which the Tanzanian administration of the time was engaging in a crackdown on public sector corruption and private sector tax evasion which created suspicion amongst some potential participants about why we wanted to know about the impact of HIV on their organisations, not all those approached agreed to be interviewed [37]. The final sample was comprised of 9 private sector business owners, 7 regional/zonal managers of private sector companies, and 7 public institution regional/zonal managers. This mixed sample of both private and public sector managers and business owners enabled the research team to capture a wide range of experiences from senior managers and owners who may face different incentives and challenges with respect to HIV and the promotion of workplace interventions. Further, the organisations were anticipated to be large-scale employers in recognition of the fact that reviews emphasise that the impacts of HIV may be more keenly felt in larger organisations, and that there will be different pressures for medium and small enterprises. Table 1 provides background characteristics of the participants that are most relevant to the study, including which sector their organisation was in, the number of employees in their organisation, their position/role in the organisation, and their age. All participants were male, with mean age of 49.47 years.

## Data collection

In February and March 2017, we conducted 23 semi-structured interviews with the owners or regional managers of key public sector institutions and prominent private sector businesses. The research team for this project was comprised of one researcher (Author A) from the UK, and three researchers from Tanzania (authors B, C and D). All interviews were conducted by Author C (a male graduate researcher) with the other authors (Authors A, B and D) not present. Interviews were conducted in the participant's workplace according to their convenience and lasted between 30 minutes and one hour. The interviews were conducted with the support of a pre-designed interview guide (see English and Swahili versions in S1 and S2 Files) which focused on: demographic details of the participant and details of the organisation they worked for/owned; how HIV had impacted their organisation (or not); how the issue of HIV was viewed in their organisation; what services/support were provided to employees around the issue of HIV; whether their organisation had a HIV policy and what was included in this policy; what types of formal workplace programmes had been implemented in their organisation; and other ways their organisation had responded to the issue of HIV. All interviews were conducted in Swahili and were recorded and subsequently transcribed and translated by NIMR employees.

**Table 1. Characteristics of organisations in the sample.**

| Participant No. | Sector | UN Sector Classification | Position of Participant | Estimated No. of Employees |
|---|---|---|---|---|
| 1 | Private | Manufacture of food products Division 10 | Owner | 93 |
| 2 | Private | Wholesale trade Division 46 | Owner | 48 |
| 3 | Public | Public administration and defence; compulsory social security Division 84 | Manager | 180 |
| 4 | Private | Financial and insurance activities Division 64 | Director | Not given |
| 5 | Private | Administrative and support service activities Division 80 | Manager | 570 |
| 6 | Private | Administrative and support service activities Division 80/ Manufacture of food products Division 10 | Owner | 380 |
| 7 | Private | Administrative and support service activities Division 80/ Transportation and Storage Division 49 | Owner | 196 |
| 8 | Public | Public administration and defence; compulsory social security Division 84 | Manager | 80 |
| 9 | Private | Information and Communication Division 61 | Manager | 98 |
| 10 | Private | Manufacturing Division 13/ Accommodation and Food Service Activities Division 55 | Owner | 200 |
| 11 | Public | Public administration and defence; compulsory social security Division 84 | Manager | 1,094 |
| 12 | Public | Public administration and defence; compulsory social security Division 84 | Director | 300 |
| 13 | Private | Information and Communication Division 61 | Director | 62 |
| 14 | Private | Financial and insurance activities Division 64 | Manager | 350 |
| 15 | Private | Construction Division 41 and 42 | Owner | 60 permanent, 90 temporary |
| 16 | Private | Construction Division 41 and 42 | Owner | 26 |
| 17 | Public | Public administration and defence; compulsory social security Division 84 | Manager | 60 |
| 18 | Private | Manufacturing Division 11 | Manager | 250 |
| 19 | Public | Public administration and defence; compulsory social security Division 84 | Director | 80 at head office, 500–5,000 at project sites |
| 20 | Private | Wholesale trade Division 47/ Accommodation and Food Service Activities Division 55 | Owner | 63 |
| 21 | Private | Manufacturing Division 11 | Manager | 150 |
| 22 | Private | Accommodation and Food Service Activities Division 55 | Owner | 50 |
| 23 | Public | Public administration and defence; compulsory social security Division 84 | Manager | 205 |

## Data analysis

The data was analysed using a thematic approach. Drawing on best practice guidance [38], during the familiarisation process, the research team conducted debriefs after every interview to discuss and document emerging themes. These were triangulated with existing conceptual frameworks [14, 17] and the author's background review [23]. Once data collection was completed, all translated transcriptions were imported into Nvivo v11. To support the initial generation of codes a coding framework was developed, reflecting emerging themes, existing conceptual frameworks and the core interview questions which included the following opening questions and follow up probing questions (see S1 and S2 Files): Has HIV ever been an issue that has had an impact on your business/organisation/company?; Do you have a HIV policy in your business/organisation?; Do you have any programmes on HIV at your business/ organisation?; Do you support your employees to help them access ARV's and treatment? In what ways?; Do you provide HIV testing for your employees? If yes, what are the views of your employees about this?. The initial coding framework included categories for 'impact of HIV', 'employer responses', 'employee issues', 'formal policy', 'formal programmes' 'workplace HIV activities' and 'shifting the burden'. The interviews were coded by author A, and peer reviewed by authors B and C to ensure the validity of codes and to develop consensus within the team.

### Ethics statement

Ethical approval was granted by the Medical Research Coordinating Committee (MRCC) of the National Institute for Medical Research, Tanzania (NIMR/HQ/R.8a/Vol. IX/2376) and also by the University of Northampton (UK) Ethics Committee (04/05/16). Further, written permission to conduct the fieldwork was obtained from the Mwanza Regional Medical Officer, the Regional Administrative Officer, Mwanza, and the District Medical Officers of Ilemala and Nyamagana districts (the districts in which the fieldwork was conducted) following standard ethical procedures in Mwanza region. All participants were provided with a project information sheet and consent form which outlined the composition of the research team, the main objectives of the study, the themes and question that would be covered in the interview, how long the interview would take, how their data would be used, the request for the interview to be recorded, and their rights to withdraw from the study. Participants provided oral and written informed consent prior to the interview.

## Results

### HIV not generally seen as an issue

The majority of employers (14/23) reported that HIV was not generally an issue for them or it was a relatively minor one, and eight employers reported never having encountered a HIV positive employee in their organisation This was explained in a number of different ways. For example, HIV is now viewed as more of a personal issue and not necessarily something that employers should get involved in, a position that is in contrast to the literature reviewed above [4, 13, 14].

> "*like I've said, it's a private issue, but even that [prevalence] rate of individuals here, it's not that big, so I can say that as an institution, I see it doesn't have a very big impact*" (Zonal Manager, Private Sector)

They also questioned whether employees would disclose this sort of information, highlighting potential conflicts of interest and understandable hesitancy on the part of employees trusting those in a position of power over them with sensitive information:

> "*they are many so you could find that among them there are people who have that problem, but they are never open. . . maybe someone does not want to be known that they have a problem so you could find that they think if we find out that they are infected we will fire them*" (Owner, Private Sector)

Employers also noted that improvements in education and access to treatment means that employees do not necessarily have to disclose their status which might explain why they had not encountered an HIV positive employee:

> "*for now many citizen workers and residents know how to live with such infections from HIV/AIDS in the sense that they either eat well or take those pills that maintains their healthy condition. . . so especially for these years, you can find that perhaps one has it and they don't know, but their health is good and they work just fine without any problems*" (Manager, Public Sector)

Most employers reported knowing of only a small number of HIV positive employees in their organisation, with only one out of the 23 participants reporting high numbers of HIV-related deaths amongst their workforce:

"*this disease started in the year eighty-three if I am not mistaken, a lot of workers have lost their lives in our department, very many people*" (Officer, Public Sector)

Reflecting this reference to the 1980s, other employers also noted that HIV used to be more of a problem in the past when mortality rates were higher and treatment far less available, comparing the present with the situation in the 1990s.

"*Different from the nineties to two thousands, important people in the company were infected with HIV, but now it is not there because people have known prevention tools, so we don't find the cases of people going for treatment and fail to come in the job*" (Manager, Private Sector)

To some extent this statement reflects the general age range of our sample, a mean age of 49.47 years, who were able to remember the devastation that the epidemic caused during the initial onset and spread prior to the recent improvements in education, prevention, and availability of effective treatment.

Finally, the type of employees that different organisations attracted was also a key factor. Some employers noted that their workforce was to some extent transient, reflecting short term contracts or high turnover rates:

"*But in my businesses I have never met such things, workers are coming and go after sometimes, others come, so I don't have permanent workers, who stay for long time*" (Owner, Private Sector)

This contrasts with other employers that noted their employees were 'elite' employees that were already aware of HIV, and so this was an issue that did not require attention:

"*Most of our workers are elites, and they pretty well understand the issue of HIV/AIDS as it is well understood now, so I don't expect in the future days because we haven't seen its effects directly which can scare, our performance at work*" (Manager, Public Sector)

### Formal policies but few formal workplace programmes

When asked whether they had a formal HIV policy in their organisation, many employers replied to the affirmative. Reflecting different governance regimes discussed below, all seven public sector employers reported formal HIV policies within their organisations, whereas five private sector employers reported not having a formal HIV policy, and two reported a formal HIV policy was in preparation. When the employers were probed further about the nature of the formal policies, it was clear that the policy was essentially an organisational policy of non-discrimination of those living with HIV that reflected national policies, rather than anything more extensive around the rights of HIV positive workers and responsibilities of employers:

"*[laughter] in short I have not gone deep on our policy but it has just described . . .treatment of workers who are HIV positive, therefore as a company it highly avoids any kind of discrimination or abusive language which can cause psychological effects to a HIV positive person, and it believes that both HIV positive and HIV negative are human being and have right to life and right to work. Therefore even if you are a leader of this company you must respect this.*" (Manager, Private Sector)

However, it was clear from the responses from public sector employers that there were different regulatory/governance frameworks regarding HIV policies, which in the public sector entailed formal, standardised procedures for dealing with HIV positive employees (assuming that they made themselves known) which included the provision of food supplements and salary uplift:

*"For those whom we have identified, we give them assistance items of food. This is to help them not to have diet problems. That is in addition to the salaries they get, we also have a top up fund which we give them quarterly. . . As an institution this is all aimed at making them realize that we still do value them and are interested in having them still work for us" (Manager, Public Sector)*

## Reliance on informal/ad-hoc workplace activities to address HIV

Whilst employers did not report any systematic, formal workplace HIV programmes, there were a range of informal, ad-hoc HIV-related activities and support mechanisms provided. These included talks about HIV by those in leadership positions, awareness events and seminars, condom distribution and some voluntary testing programmes. Employers in the private sector also reported having life and/or medical insurance for their workers or that they had provided financial support to the families of workers who had died from AIDS, but this was by no means the norm across the sample. Often employers reported that HIV activities were conducted by local NGOs or invited medical professionals, rather than these being in-house activities:

*"I think sometime last year there was an NGO here and volunteered though I don't have enough data, because the event took place at the headquarters, but there was also a certain NGO which was supplying condoms in our offices and even in areas like universities and we agreed to that as it is one of the way towards fighting for this disease, so we were even putting different facilities in the washrooms, private areas where one could easily access it" (Zonal Manager, Private Sector)*

*"But also a certain year, ee I remember [name of employer], they tested all workers all over the country, this was done by the people from AMREF" (Director, Private Sector)*

There were also examples of HIV activities being conducted by local health professionals who had a relationship with specific employers:

*"There is one sister, we were giving her a chance after every three months to have conversations and advice the workers, She is called [healthcare professionals name] she was at Bugando in HIV department" (Owner, Private Sector)*

Regular meetings, either monthly or weekly, were also used to discuss HIV and other health issues with the workforce:

*"Other thing that we are doing in our monthly meeting, we invite the doctors from AR to provide trainings, and encouragements, education about different health issues at work including HIV/AIDS" (Director, Public Sector)*

These findings suggested that whilst employers did not feel the compulsion to introduce formal workplace programmes, they were numerous examples of engagement, often reflecting

the personal interests linked to previous experiences, as noted in previous studies (Dickinson and Stevens 2005), of those in leadership positions:

> "*So you find that I use the same lesson to teach these people, so there are a lot of things that push me but firstly it is the risk I overcame after a few of my relatives whom I heavily depended on to die because of AIDS.*" (Owner, Private Sector)

This is further evidence of the patchy and limited response by employers too HIV.

## Compulsory HIV programmes

There were, however, more formalised HIV-related arrangements for firms working in specific sectors such as food production. These sectors were either seen as requiring frequent HIV-testing for health and safety reasons, or that were linked to patterns of transmission of the virus. In two food production companies, participants reported that government regulations enforce regular testing programmes for employees:

> "*We always test the workers after every six months It's compulsory, that's on among the laws that we are given by TFDA [Tanzania Food and Drug Authority] and the concerned authority* "(Owner, Private Sector)

Therefore, employers included this in company contracts: *"Since we started the company we've been having that custom because it's among the criteria that make us run this business, That that's our mandatory. So it's a must when they sign our contract, it's one important thing which is there that after every six months, we do medical checkup"* (Owner, Private Sector)

However, subsequent checks with government agencies suggested that the HIV testing component of these regular health checks that food workers undergo is not compulsory due to regulations (and therefore not legal). We have been unable to access written documentation of the relevant policy that was in force at the time of fieldwork, so are unable to assess the extent to which employers had misunderstood these regulations or were deliberately contravening them.

However, there are significant consequences of compulsory testing programmes, with employers in this sector reporting that employees did not turn up for the test or refused to test and thus lost their employment:

> "*So it was compulsory for the industries to be inspected because we deal with food issues so we were forced to take workers for test. So I also took the doctors at the industry and they emphasized on testing otherwise if you don't want then you have to resign. But still some people had an excuse and left the day before doctors came claiming that they didn't want to see the doctors*". (Owner, Private Sector)

> "*So we tell them that once they refuse, we don't give them contracts. And if we won't give you contracts you won't be with us, you will be forced to search for another place that doesn't have the system of producing food*" (Owner, Private Sector)

This illustrates one of the challenges of compulsory testing programmes and the impact that may have for those who either already know they are HIV positive or suspect they are. It also emphasises the enhanced vulnerability that workers who are HIV positive face when trying to access formal employment in a context where this still accounts for a low proportion of total employment (ILO 2018).

In another sector related to infrastructure, Tanroads, the government agency that is responsible for managing the tender process for infrastructure related projects, has a formal policy (TANROADS 2018) that insists on a proportion of the contract being spent on HIV related activities for both workers and local communities around infrastructure projects:

"*we are very grateful to Tanroads because before we start their assignments they put aside a fund, they clearly put a separate fund on the bill of quantities, you see it very clearly that this is for HIV awareness and so we normally go with a team from the health sector and also the contractors themselves and yourself it is now a law that has been put in place and you go to the places where you have your ongoing projects together with your employees.*" *(Owner, Private Sector)*

The policy reflects a range of issues, such as the number of migrant workers attracted to the sector, temporary workforce mobility associated with infrastructure projects, and high rates of HIV transmission amongst communities located close to main roads [39, 40].

## Supportive workplace initiatives

Alongside the compulsory HIV programmes, some employers reported ways in which they had previously managed HIV-positive employees, and which were sensitive to their employees' HIV status and the working practices required of their role. These approaches aimed at providing a supportive working environment that would also ensure continued access to HIV related services:

"*I do not want them to be posted at place that is a bit cold but if there is somewhere warmer like during the afternoon or customers that I know provides food for the guards I would give them the priority because I know they will get food. They need the food because of the medication they are taking. Because there are some customers who treat my workers like family so when they guard there they are given breakfast and if it is dinner, they are given. So I give priority to those who have brought me their results, because I know their condition I can decide where they should be arranged*" *(Owner, Private Sector)*

"*We have health insurance for workers, therefore when a person has disclosed we care for them and most times we put him/her in the environment that is comfortable for him/her. The problem is when s/he don't disclose because our company does not put a person in one place for long time, we have the system of rotating after some times, therefore if a person has not disclosed we can allocate him/her in the environment where s/he will miss the clinic. You know there are rural market environments, if a person will go there it is difficult to get services but if s/he has disclosed we normally allocate him/her in town where s/he can get the services easily.*" *(Manager, Private Sector)*

These responses involve some reorganisation of rotas and appointments, but importantly did not necessarily involve economics costs on the part of the employer. The work required by their organisation is still performed, but in a slightly changed configuration.

Other employers, both private and public, also noted that it would also be possible to redeploy a worker to another part of the business:

"*you can find them work that doesn't associate with food. There's a cleaning unit here, we have outdoor work, procurement so you tell them maybe they will deal with shopping like*

*buying soap and what not, so we shift you from your unit simply because you were working in a unit that is sensitive (Owner, Private Sector3)*

"*Those who have been infected there are special programs for taking care of them including giving them nutrition and other things that would help them in living, if a person was in the heavy works part to find him/her soft work which is easy according to the status s/he has at that time*" (Manager, Public Sector)

Finally, there was also support for employees being able to go to the clinic by scheduling their days off at the same time:

"*And also when they have started treatment I follow up on their schedule of attending the clinic. So if they go to get medicine twice a month I put it on the days they are supposed to be having an off day. So I would tell them not to come on their off day so they can go and get their medicine, that way other workers just think they are off*" (Owner, Private Sector)

These supportive responses do, of course, rely on workers being comfortable enough to disclose their HIV status to their employers. As noted above, this is not always the case, and so the potential for responses of this nature will be heavily influenced by the nature of specific employer-employee relationships.

## Shifting the burden?

In contrast to these positive narratives, employers did not report many examples of shifting the burden onto either the state or HIV positive employees. However, three employers reported that they conducted pre-employment testing, though this was framed as a requirement that would enable them to provide the right support or for health insurance purposes, rather than a means of avoiding hiring HIV positive employees:

"*Fortunately I have never had such an employees at my level. But even when we employ them, we conduct a test on them before employment. But even if it happens that there is someone who is infected with HIV, we will still employ him, because it's their right, if there will be further problems ahead then he will get treatment just like the others*" (Manager, Private Sector)

Additionally, when asked whether their organisation had implanted any testing or treatment programmes, one employer responded with the view that this was something that should be provided by the state:

"*We as an institution we have not reached those levels of involving ourselves with their medications and treatment. We have always held the view that their treatments is nationally planned by the government for each one of them*" (Director, Public Sector)

Whilst this is not necessarily an active shifting of the burden onto the state, it does suggest that employers do not always think in the way anticipated by proponents of the business-case approach.

## Discussion

Our findings illustrate the mixed and diverse ways that employers have responded to the epidemic, and the multiple dimensions which influence the response which include sector-specific requirements, formal national and company policies, the nature of different workforces

across sectors, the personal interests of some employers with respect to HIV, and employer-employee relations [20, 23, 28, 29]. Echoing themes reflected in the broader literature, the response is limited and variable across different organisations, with little evidence of formal workplace programmes of the type championed by the ILO being implemented [20, 21]. Where organisations have formal HIV policies these primarily reflect standard national non-discrimination policies, rather than more substantive or holistic approaches. As important as the issue of non-discrimination is, this revealed the limited nature of formal HIV policies and the relative lack or concern about HIV. No employers reported that they had a formal HIV workplace programme of the sort implemented in the mining sector in South Africa [19], and so HIV-related activities conducted within the workplace are primarily ad-hoc interventions that often involve actors from the third sector. There was also evidence of some positive responses that diverged from the formal workplace programme approach advocated for in the literature [15], which entailed employers re-organising productive activities and rotas to support HIV positive employees.

The nature of this limited response can in part be explained by the fact that for many employers, HIV is simply not a significant concern. Advances in the availability of testing and treatment in urban areas and the rollout of free ARVs in Tanzania [41] have made it possible for HIV positive workers to access treatment and keep their diagnosis private. Employees who have stable employment are also in an enhanced position to adhere to treatment regimes and continue to be as productive as other workers. Further, the lower background prevalence rate of around 6.8% in Mwanza region in comparison to Southern (and especially South) Africa (where the momentum for workplace programmes derived from) make it highly likely that HIV prevalence rates within organisations (subject to sectoral variations) are also lower. These factors significantly reduce the 'costs' of HIV to employers, and so across our sample there is little evidence of employers viewing this issue through the economic lens of the cost-benefit approach.

With HIV not seen as a major issue (though still an issue requiring some attention), responses are driven by a range of other factors, such as internal advocates in leadership positions, terms of government tenders, health and safety legislation for specific sectors, conditions of some private insurance schemes and relationships with NGOs and other local contacts who provide ad-hoc HIV services. Our findings thus challenge the extent to which lessons from case studies drawn from advanced sectors in high prevalence settings can be generalised across different contexts in sub-Saharan Africa. Attempts by policy makers to encourage businesses to 'do the right thing' by persuading them that this also makes good business sense [8, 9, 11, 14, 15] are unlikely to gain much traction in many contexts across the continent which experience similar or lower prevalence rates, and where public testing and treatment services/programmes have been successfully implemented.

The evidence presented in this study also casts doubt on whether the workplace is an appropriate or effective site for HIV intervention efforts. Whilst there were a mixture of ad-hoc and more formalised responses reported by our participants that suggests there can be a role for employers, in lieu of formal standards and expectations (especially in the private sector), coverage will continue to remain variable across different workplaces and sectors [21, 23]. Whilst it may not be fruitful to actively discourage employers from implementing HIV-related activities, it has also been emphasised that in some cases, compulsory testing programmes resulted in some employees who may have been HIV positive or suspected they were not turning up for the tests and thus losing their employment. This unintended consequence, reflecting concerns on the part of the employees about revealing a potential HIV diagnosis to their employer [42] illustrates that rather than supporting HIV positive workers in the workplace, some workplace interventions act as an additional barrier for workers to maintain access to employment

in a context in which secure employment is highly sought after. This may also help to explain why some employers had not come across HIV positive employees, as the option of leaving work may be more palatable than disclosing to an employer.

This emphasises the clear tension identified across the responses from employers between workers hiding or not wanting employers to know their status, and the reported positive and supportive role that employers can play. The creative ways in which roles and responsibilities are re-organised in some organisations to enable employees to both work and access food and/ or HIV services may comprise a different 'win-win scenario' for employers and HIV positive workers in comparison to the framing decisions through the business case approach. This alternative win-win scenario is embedded in an understanding of the labour process and how that can be reconfigured to accommodate HIV positive workers, supporting HIV positive employees to maintain access to employment and sustain productivity at little cost to employers. However, responses of this nature rely on transparent and trustworthy relationships between employers and employees which are often fragile given the power that employers have over employees, as well as requiring productive activities that can be reconfigured to accommodate the needs of HIV positive employees. These unequal power relations, which in some contexts have been exacerbated by HIV [27] present a challenge for any workplace-based approach to HIV, with employers claiming they are unable to support employees if they don't know their status, but employees understandably reluctant to forward this information. Whilst workplace programmes will undoubtedly work well for some employees, for others this will be something that may put them in a more vulnerable position and are to be avoided.

A final issue to reflect on is the critical perspective that accuses employers of 'shifting the burden' [25] of HIV onto the state and households who have a member living with HIV. Whilst there was some evidence of pre-employment screening programmes which in the past have been used to help employers avoid hiring HIV positive workers [24–26], there is clearly more research needed in this area. Our participants primarily framed pre-employment screening as a process which would enable employers to provide additional support to workers or for private health insurance schemes that would cover treatment. These narratives must be treated with caution, as despite formal national polices that prohibit discrimination (in the workplace) based on HIV status and that were referred to frequently by employers, evidence suggests that PLWHIV experience discrimination in the workplace, including being refused a work or employment opportunity, losing a job or source of income, and being refused promotion and/ or having their job description changed due to their HIV status [24, 43, 44]. Whilst the experience of HIV positive workers will undoubtedly vary across workplaces [45], for a clearer picture to emerge the voices of the HIV positive population (both employed and unemployed) need to be incorporated into the analysis.

Our findings have several implications for policy makers in Tanzania which will also be applicable to other contexts in sub-Saharan Africa. Firstly, if employers are to be encouraged to engage with HIV, workplace activities should be concentrated on information campaigns, condom promotion and/or distribution and the distribution of self-testing kits [46, 47]. These interventions can support government efforts, and they also enable workers to maintain autonomy and confidentiality over HIV-related matters. In contrast, workplace treatment and testing programmes are likely to have adverse impacts for HIV positive employees, and this is especially pertinent with regards to compulsory testing programmes forced on employees which frequently end in employees losing employment. Secondly, policy makers should not rely on employers to complement government efforts to expand access to testing and treatment–there is no substitute for funding these services through the public health system, and the economic imperative for them to do so is weak. Where services such a voluntary testing programmes are provided, local and national NGOs or external medical providers are more

appropriate organising agents than employers, and will have more ability to safeguard the confidentiality of employees who have concerns about HIV, including regarding their serostatus [4, 48]. Finally, government agencies responsible for workplace health and safety in the food production sector should reinforce to employers that HIV testing should not be included in any compulsory routine health testing that workers undergo.

More generally, whilst the workplace has been promoted as an ideal and even optimal site for health promotion in low and middle income countries [49], our findings provide a clear example of the complexities and challenges of successful interventions. This is all the more pertinent given the variable evidence on the effectiveness of workplace programmes that have addressed a wide range of health issues, and in some cases the reversal or worsening of core outcomes [49]. Careful a priori consideration of the unintended consequences, employee concerns and the nature of employer-employee relationships may help to improve their effectiveness. However, policy makers must also not lose sight of public health system successes–in this study, the irony is that successful public programmes to expand access to testing and treatment have undermined an expanded role for employers which is predicated on a presumption that states have limited capacity to do so.

Our study has several limitations and potential biases. Firstly, given the nature of the research topic it is possible that employers who did not adhere to national HIV policy guidelines may have declined to participate. This has the potential for the responses of employers to be reported in an overly positive and supportive way. Related to this, given the unequal power relations between our participants and the author who conducted the interviews, there may have been times when our participants successfully resisted follow up questions or probes on sensitive issues, and may have wanted to present themselves in a positive light to an interviewer from a local government research organisation. A further limitation is that in cases such as the reporting of compulsory testing in the food production sector, we were unable to access documentary evidence of the policy in force at the time of our fieldwork.

Another important limitation is that our research focuses on employers and their views about how employees experience HIV in the workplace. We are aware that employees may have a different opinion on this, and so future research should include their perspectives to enable the triangulation of employer and employee views. Having said that, the views of employers, as powerful and influential figures in their organisations, constitutes a valid perspective to report independently of employee's views and experiences. Finally, whilst we report a range of valuable insights, our sample size is not large enough for us to draw conclusions about sectoral differences, experiences of organisations with different numbers of employees, or differences between private and public sector organisations (though some of our results touch upon these issues) in terms of how employers respond to the issue of HIV.

## Conclusion

This study has enabled a critical reflection on the business case approach that has been used to promote the expansion of workplace programmes. Our findings highlight in general that employers do not see HIV as a serious issue, in part due to the enhanced availability of testing and treatment services and the tension embedded in employer-employee relations. The majority of organisations, including all public sector organisations, have formal HIV policies, but this does not translate to the implementation of formal workplace programmes of the kind promoted by the UN/ILO. Instead, employers provide a wide range of ad-hoc HIV interventions, including the provision of condoms, HIV awareness and education seminars, and some testing programmes, primarily conducted by external organisations. We also do not find much evidence of employers making decisions in the way that the business case approach

suggests. Whilst workplaces can be appropriate sites for some interventions, the unintended consequences of compulsory testing programmes where workers can lose access to employment suggests that testing and treatment programmes should remain out of scope to protect the confidentiality and employment status of workers who are HIV positive. Future research on this issue must include the voices of PLWHIV to fully understand their experiences of living with, and working with, HIV, as well as how regulatory regimes and state-employer relations shape the dynamics of employer responses.

## Supporting information

**S1 Checklist. COREQ checklist.**
(PDF)

**S1 File. IDI researchers guide EmpResp ENG REV2 final: Interview guide in English.**
(DOCX)

**S2 File. IDI researchers guide EmpResp SWHL REV2 final: Interview guide in Swahili.**
(DOC)

## Acknowledgments

Institutional and logistical support was provided by the National Institute for Medical Research, Mwanza Branch, Tanzania. Joel Bigesso helped with introductions and invitations for potential research participants.

## Author Contributions

**Conceptualization:** Kevin Deane, Joyce Wamoyi.

**Formal analysis:** Kevin Deane, Joyce Wamoyi.

**Funding acquisition:** Kevin Deane, Joyce Wamoyi.

**Investigation:** Samwel Mgunga.

**Methodology:** Kevin Deane, Joyce Wamoyi, Samwel Mgunga.

**Supervision:** John Changalucha.

**Writing – original draft:** Kevin Deane.

**Writing – review & editing:** Joyce Wamoyi, Samwel Mgunga, John Changalucha.

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
