## [Decision Letter · Decision Letter 0]

31 Aug 2021

 PGPH-D-21-00354 A win-win scenario? Employers’ responses to HIV in sub-Saharan Africa: qualitative evidence from Tanzania PLOS Global Public Health

Dear Dr. Deane,

Thank you for submitting your manuscript to PLOS Global Public Health. After careful consideration, we feel that it has merit but does not fully meet PLOS Global Public Health’s publication criteria as it currently stands. Therefore, we invite you to submit a revised version of the manuscript that addresses the points raised during the review process. 

There are a number of areas where the reviewers believe revisions are necessary to improve the quality of the manuscript. In this regard, the authors are advised to incorporate the feedback from the reviewers into a revised version of the manuscript. For the purpose of providing the recommended reporting requirements for a qualitative study, the revised manuscript should be returned with the completed Standards for Reporting Qualitative Research (SRQR) checklist (https://www.equator-network.org/reporting-guidelines/srqr/) indicating the page and line or paragraph numbers for each reporting criteria. Please see the additional editor feedback for recommended revisions at the end of this decision letter.

We look forward to receiving your revised manuscript.

Kind regards,

Patrick A. Palmieri, DHSc, DPhil(Hon), EdS, MBA, MSN, PGDip(Oxon), ACNP, RN, CPHRM, CPHQ, FAAN

Academic Editor

Journal Requirements:

Additional Editor Comments (if provided):

Specific to the methods section, I highly recommend a more structure approach to the section to include headings for study design (as one is not stated in the manuscript), setting and participants (including the sampling technique), data collection (recording and transcription were not detailed), ethical approval (specific protocol numbers not provided), data analysis (process for analysis was not described), and trustworthiness (elements were not discussed). In addition, the introduction requires the last paragraph to provide the purpose of the research rather than a discussion about the findings of the study. Furthermore, the discussion section requires additional information about the global context of the work in relationship to the findings of the current study. Also, the conclusion seems to include more discussion rather than a summary of the key points to take away from the study. Finally, the reference need to be updated as 22 of the 30 were published prior to 2015, with more than half of these prior to 2010.

Reviewers' comments:

Reviewer's Responses to Questions

**Comments to the Author**

1. Does this manuscript meet PLOS Global Public Health’s publication criteria? Is the manuscript technically sound, and do the data support the conclusions? The manuscript must describe methodologically and ethically rigorous research with conclusions that are appropriately drawn based on the data presented.

Reviewer #1: No

Reviewer #2: Partly

2. Has the statistical analysis been performed appropriately and rigorously?

Reviewer #1: Yes

Reviewer #2: N/A

3. Have the authors made all data underlying the findings in their manuscript fully available (please refer to the Data Availability Statement at the start of the manuscript PDF file)?

Reviewer #1: No

Reviewer #2: No

4. Is the manuscript presented in an intelligible fashion and written in standard English?

Reviewer #1: No

Reviewer #2: Yes

5. Review Comments to the Author

Reviewer #1: Reviewer's report on the article titled “A win-win scenario? Employers’ responses to HIV in sub-Saharan Africa: qualitative evidence from Tanzania”

Date: 4th August 2021

General Comment: The authors are addressing an important subject particularly within the context of a developing country like Tanzania where about 4.9% of the active population (aged 15 years and above) live with HIV and that, discriminatory attitudes towards people living with HIV are still prevalent. However, the manuscript has one major limitation related to the design of the study as detailed in paragraph 1.

1. Sub-Saharan African (SSA) region has more than 40 countries. Accordingly, the findings and associated conclusions and policy implications presented in this manuscript do not offer an opportunity to understand how HIV is addressed in workplaces in SSA. Institutional or workplace responses to HIV are largely dependent on national policies and strategies on HIV prevention and treatment that are in place, which are likely to differ from one country to another. Evidence from a multi-country study design would have added value to the present study. For example, the authors state in lines 19 and 20 that “Our findings suggest that HIV is not generally a big issue, and hence HIV interventions are primarily ad-hoc with few formal HIV workplace programmes”. This is not likely to be case in other countries in SSA. Indeed, the authors point out in lines 75 – 79 that “The empirical evidence, which primarily originates from a narrow range of multinational/blue chip firms in South Africa (11), is less compelling once evidence in other sectors and countries is accounted for, and especially with respect to whether workplace programmes have the intended impact of reducing new HIV infections and HIV-related morbidity and mortality”.

2. It is not evident as to why Tanzania in general and Mwanza city in particular was considered as a representative case study among the several countries in SSA and cities in Tanzania, respectively. This should be clarified.

3. It is also not evident when the data collection took place and for how many days.

4. Line 102: The authors conducted 23 semi-structured interviews. For clarity, were there pre-formulated set of interview guide questions? If yes, what informed the design of the questions? In which language (English or Kiswahili) were the interviews conducted?

5. Lines 139 – 143: For clarity, consider revising the sentence “This approach (see for a full discussion of the strengths and limitations of this approach see (30) reflected the potentially sensitive topic under discussion, and in light of the linguistic limitations of one of the authors (Author A), ensured that interviews could flow without the continuous interruptions necessitated if this had been done using the research assistant or other authors as interpreters”.

6. Line 145: State the ethical clearance certificate number.

7. Lines 147 – 148: Which authority of local government and community issued the permission to conduct the study considering the fact that the study was not conducted at the communities? Please clarify.

8. Lines 148 – 149: “…governmental and community leaders and all participants gave oral and written consent prior to the interview” Why was it necessary that all participants give oral and written consents?

9. Lines 148 – 149: What was the content of the written consent form?

10. Lines 151 - 187: It would be more informative if the authors could first present in a tabular form, characteristics of the respondents in terms of (among other attributes) education, sex, age, years in the current position, and marital status.

11. Lines 152 – 154: The authors stated that “The majority of employers reported that HIV was not generally an issue for them or a relatively minor one. Some viewed it as more of a personal issue and not necessarily something that employers should get involved in, a position that is in contrast to the literature reviewed above” For clarity, the words ‘majority’ and ‘some’ need to be quantified. Further, for ease of reference, the authors could mention specific literature instead of stating the “literature reviewed above”.

12. Line 155 – 156: Improve the quote for clarity. For example, the first and second ‘its’ in line 155 need to be changed to ‘it’s’ Further, the word ‘issues’ need to be changed to ‘issue’.

13. Line 163: Quantify ‘some employers’ for clarity.

14. Line 166: Change the word ‘life’ to ‘lives’.

15. Line 168: Quantify ‘other employers’ for clarity.

16. Lines 174 – 175: The statement that “… as wealthier men in the Tanzanian context are also often likely to be older and thus able to remember…” is misrepresentative in the sense that first, it is based on remarks from one respondent and second, the sample is not representative of all wealthier men in Tanzania. Further, it is important to be aware that income, education and occupation are three interrelated factors, but have different policy implications! Highly educated individuals are generally in the highest wealth quintiles than their less educated counterparts ceteris paribus.

17. Lines 178 – 179: Quantify ‘other employers’ for clarity

18. Line 183: Quantify ‘other employers’ for clarity since the quote in lines 185 – 187 does not reflect many employers.

19. Line 189: Quantify ‘many employers’ for clarity.

20. Line 194: Add an ‘s’ to the word ‘avoid’.

21. Line 200 – 225: The findings presented here are rather surprising since prevention against the HIV/AIDS pandemic is one of the objectives set annually, virtually in every public institution, with budget allotted to implement the set target in the context of Tanzania. Virtually all quotes are from private sector. A comparative analysis of the situation in the private vs public workplaces would add value to the findings.

22. Line 238 and 239: Change ‘its’ after the word ‘because’ and ‘contract,’ to ‘it’s’.

23. Line 273: Consider revising the text “… but if it somewhere…” for clarity.

24. Lines 280 – 288: Consider improving the grammar for clarity.

25. Lines 314 – 317: Consider improving the grammar for clarity.

26. Line 323: Incomplete sentence.

27. Line 324 – 325: The statement that “… it does suggest an attempt to shift the responsibility” need to be rephrased as it it does not accurately reflect what the government of has always been advocating since the first case of HIV occurred in Tanzania. That is, voluntary counselling and testing. In this case, the response of the respondent in lines 319 and 320 is correct. That is, “…one employer responded with the view that this was something that should be provided by the state”.

28. Lines 326 – 433: The discussion of the findings is generally weak. Consider restructuring the discussion such that each paragraph reflects particular finding of the study and support the discussion by existing literature from studies conducted in similar settings elsewhere. In most cases, the authors have discussed the findings of the study without accompanying evidence from other similar studies. This limits the validity of the discussions and hence, the findings presented.

29. Line 333: Put a full-stop after the second bracket and a comma after the word ‘policies’.

30. Line 343: The statement “…primarily ad-hoc interventions that often involve actors from the third sector” need to be rephrased considering the fact that there are formal and structured HIV/AIDS programmes in public settings in the context of Tanzania.

31. Line 345: Provide evidence to support the statement “… advocated for in the literature”.

32. Line 350: Qualify the statement “…in comparison to Southern Africa”.

33. Lines 355 – 356: The argument that “…having 5 HIV positive employees out of a 100 may not result in a high economic burden for employers of the magnitude reported in the mining sector in South Africa” is not supported by empirical evidence. Depending on the skills and experience, even one (1) employee, irrespective of the number of other remaining, can paralyses or cause significant disruptions of the services of an organization if the said employee is incapacitated by a health shock. I recommend that the authors reexamine the said argument.

34. Line 376: Better use ‘in this study’ instead of ‘above’ in the statement “The evidence presented above”.

35. Line 423 – 425: The policy implication presented is largely not supported by findings from the study. It did not come out explicitly from the findings of the study that what is happening in the study areas is due to limited information or limited distribution of condoms or these would potentially reduce or lessen the negative attitudes towards PLWHIV. If information campaigns are necessary, it would be more informative to provide examples (informed by the findings of the present study) of key message to consider during the implementation of the information campaigns in order to achieve the intended purposes. Further, in order for the proposed intervention of distribution of self-testing kits to be meaningful, it would be informative if the authors could provide evidence of the contexts where such an intervention has been found to be useful in relation to the findings of the present study.

36. Line 431: Could the authors cite examples of a local and a national NGO for clarity?

37. Lines 432 – 433: There is need to provide evidence to support the argument that engagement of NGOs than employers is more likely to “safeguard the confidentiality of employees who have concerns about HIV, including regarding their serostatus”.

38. Line 439: Change the word ‘has’ after the word ‘that’ to ‘have’ to reflect that the statement is referring to plural (programmes and HIV policies). Further, it would be more informative to cite the referred “literature” (“…reported in the literature”) at the end of the word ‘literature’.

39. Lines 442 – 444: It is inappropriate to generalize that the workplace is not an “appropriate site of intervention due to inevitable inequities in employer-employee power relations that can increase the vulnerability of HIV-positive workers, as well as the unintended consequences of compulsory testing programmes” since in the context of Tanzania, the public service is guided by the “Standing Orders for the Public Service” that provides what public servants should do at the workplace in all work-related undertakings. It is therefore, not expected that employer-employee power relations would influence decisions in human capital development including HIV/AIDS intervention programmes that are guided by national policies on HIV/AIDS.

40. Line 448: Change the word ‘of’ after the word ‘Institute’ to ‘for’ to read ‘National Institute for Medical Research’ instead of ‘National Institute of Medical Research’.

Reviewer #2: The manuscript is well written and presents relevant subject. The presented data were collected in 2017; however, the results are still valid in the sense that they highlight the potential role of the employers when it comes to HIV interventions.

Specific comments:

Include a reference of MRCC ethical approval. See lines 145-146

Indicate the qualitative analysis framework used. Eg. Content or Thematic analysis by who? Also describe who analyzed the data.

The presentation of the findings/results should be guided by the chosen analytical approach. For example, as a reader, I would like to know whether the subtitles are 'Themes' or 'Categories'.

NB: The authors may use Critical Appraisal Skills Program (CASP) Tool to review the methods, and results sections. This will increase clarity to the readers.

6. PLOS authors have the option to publish the peer review history of their article (what does this mean?). If published, this will include your full peer review and any attached files.

**Do you want your identity to be public for this peer review?** For information about this choice, including consent withdrawal, please see our Privacy Policy.

Reviewer #1: No

Reviewer #2: No

---

## [Decision Letter · Decision Letter 1]

8 Apr 2022

PGPH-D-21-00354R1

A win-win scenario? Employers’ responses to HIV in Tanzania: a qualitative study

Dear Dr. Deane,

Thank you for submitting your manuscript to PLOS Global Public Health. After careful consideration, we feel that it has merit but does not fully meet PLOS Global Public Health’s publication criteria as it currently stands. Therefore, we invite you to submit a minor revision of the current version of the manuscript that addresses the points raised during the review process. The manuscript was reviewed by the reviewers from the first round with an attachment provided by the feedback.

We look forward to receiving your revised manuscript.

Kind regards,

Patrick A. Palmieri, DHSc, DPhil(Hon), EdS, MBA, MSN, PGDip(Oxon), ACNP, RN, CPHRM, CPHQ, FFNMRCSI, FAAN

Academic Editor, PLOS Global Public Health

Journal Requirements:

1. Your co-authors, Joyce Wamoyi (j.wamoyi@gmail.com), Samwel Mgunga (smgunga2@gmail.com), and John Changalucha (jchangalucha@yahoo.com), have not confirmed authorship of the manuscript. We have resent them the authorship confirmation email; however please check that the above email address for them is correct and follow up personally to ensure they confirm. Please note that we cannot pass your manuscript to Production until we have received confirmations from all co-authors. 

2. Please update your Competing Interests statement. If you have no competing interests to declare, please state: “The authors have declared that no competing interests exist.”

Reviewers' comments:

Reviewer's Responses to Questions

**Comments to the Author**

1. If the authors have adequately addressed your comments raised in a previous round of review and you feel that this manuscript is now acceptable for publication, you may indicate that here to bypass the “Comments to the Author” section, enter your conflict of interest statement in the “Confidential to Editor” section, and submit your "Accept" recommendation.

Reviewer #1: (No Response)

2. Does this manuscript meet PLOS Global Public Health’s publication criteria? Is the manuscript technically sound, and do the data support the conclusions? The manuscript must describe methodologically and ethically rigorous research with conclusions that are appropriately drawn based on the data presented.

Reviewer #1: Yes

3. Has the statistical analysis been performed appropriately and rigorously?

Reviewer #1: Yes

4. Have the authors made all data underlying the findings in their manuscript fully available (please refer to the Data Availability Statement at the start of the manuscript PDF file)?

Reviewer #1: No

5. Is the manuscript presented in an intelligible fashion and written in standard English?

Reviewer #1: Yes

6. Review Comments to the Author

Reviewer #1: (No Response)

7. PLOS authors have the option to publish the peer review history of their article (what does this mean?). If published, this will include your full peer review and any attached files.

**Do you want your identity to be public for this peer review?** For information about this choice, including consent withdrawal, please see our Privacy Policy.

Reviewer #1: No

---

## [Editor Report · Decision Letter 2]

29 Aug 2022

PGPH-D-21-00354R2

A win-win scenario? Employers’ responses to HIV in Tanzania: a qualitative study

Dear Dr. Deane,

Thank you for submitting your manuscript to PLOS Global Public Health. After careful consideration, we feel that it has merit but does not fully meet PLOS Global Public Health’s publication criteria as it currently stands. Therefore, we invite you to submit a revised version of the manuscript that addresses the points raised during the review process.

The journal requests that qualitative articles include an appropriate reporting guideline, such as COREQ or SRQR, to ensure inclusion of essential elements such as clear research aims, reproducible analytic methods, and consideration of bias and limitations. Please complete these guidelines and update the manuscript accordingly - note further comments below on the key elements required to strengthen the manuscript for further consideration. 

We look forward to receiving your revised manuscript.

Kind regards,

Hannah Hogan Leslie, PhD

Academic Editor

Journal Requirements:

Additional Editor Comments (if provided):

- Please include CORE-Q or SRQR (http://journals.plos.org/globalpublichealth/s/submission-guidelines#loc-qualitative-research) checklist and note areas for additional detail to adhere to these guidelines, such as inclusion/exclusion criteria, methods of approaching potential respondents, and reasons for non-participation. Please spell out the ‘core research questions and aims’ (line 168/169) that guided the codes earlier in the manuscript, and consider using the start of the results section to describe the overall coding framework arrived at during data analysis. The discussion section must include consideration of potential bias and limitation in the study findings, and could be streamlined to consolidate interpretation of the findings and ensure they are put in context of other research and other settings.

- The Study Setting paragraph suggests Mwanza is a relatively low HIV prevalence area, although 7% general population is not necessarily low in an absolute sense but primarily in comparison to heavily affected cities/business in South Africa where prior research has focused. It would be helpful to contextualize this in terms of the HIV epidemic and in terms of other diseases.

- Finding around required testing for food service workers seems quite important — are employers misinterpreting or violating policy in requiring such testing? The statement, “However, it became clear that the HIV testing component of the regular health checks that food workers have to undergo is not compulsory due to regulations but is included in company contracts” is insufficient in presenting where this information comes from and whether in fact the checks are legal but not required vs. illegal, given that they can be considered employment discrimination against those testing positive. Please spell out further the interviewee’s apparent understanding of the law/policy in the Results and expand how it pertains to actual law in the discussion (or results if the correct application of the law was identified in the course of other interviews). From an ethical perspective, I wonder if the research team considered it necessary to report on this requirement if in fact it is not in keeping with current law

- Minor: Define 95/95/95 goals, spell out ILO on first use
---

## [Editor Report · Decision Letter 3]

20 Oct 2022

A win-win scenario? Employers’ responses to HIV in Tanzania: a qualitative study

PGPH-D-21-00354R3

Dear Dr Deane,

We are pleased to inform you that your manuscript 'A win-win scenario? Employers’ responses to HIV in Tanzania: a qualitative study' has been provisionally accepted for publication in PLOS Global Public Health. Thank you for responding to the last round of feedback.

Best regards,

Hannah Hogan Leslie, PhD

Academic Editor